# Association between Neighbourhood Deprivation Trajectories and Self-Perceived Health: Analysis of a Linked Survey and Health Administrative Data

**DOI:** 10.3390/ijerph20010486

**Published:** 2022-12-28

**Authors:** Sékou Samadoulougou, Laurence Letarte, Alexandre Lebel

**Affiliations:** 1Evaluation Platform on Obesity Prevention (PEPO), Quebec Heart and Lung Institute, Quebec, QC G1V 4G5, Canada; 2Centre for Research on Planning and Development (CRAD), Université Laval, Quebec, QC G1V 0A6, Canada

**Keywords:** self-perceived health, neighbourhood, deprivation, inequality

## Abstract

Life course exposure to neighbourhood deprivation may have a previously unstudied relationship with health disparities. This study examined the association between neighbourhood deprivation trajectories (NDTs) and poor reported self-perceived health (SPH) among Quebec’s adult population. Data of 45,990 adults with complete residential address histories from the Care-Trajectories-Enriched Data cohort, which links Canadian Community Health Survey respondents to health administrative data, were used. Accordingly, participants were categorised into nine NDTs (T1 (Privileged Stable)–T9 (Deprived Stable)). Using multivariate logistic regression, the association between trajectory groups and poor SPH was estimated. Of the participants, 10.3% (95% confidence interval [CI]: 9.9–10.8) had poor SPH status. This proportion varied considerably across NDTs: From 6.4% (95% CI: 5.7–7.2) for Privileged Stable (most advantaged) to 16.4% (95% CI: 15.0–17.8) for Deprived Stable (most disadvantaged) trajectories. After adjustment, the likelihood of reporting poor SPH was significantly higher among participants assigned to a Deprived Upward (odds ratio [OR]: 1.77; 95% CI: 1.48–2.12), Average Downward (OR: 1.75; CI: 1.08–2.84) or Deprived trajectory (OR: 1.81; CI: 1.45–2.86), compared to the Privileged trajectory. Long-term exposure to neighbourhood deprivation may be a risk factor for poor SPH. Thus, NDT measures should be considered when selecting a target population for public-health-related interventions.

## 1. Introduction

Exposure to neighbourhood deprivation may affect health predictors and outcomes [1,2,3,4]. The literature suggests associations between neighbourhood deprivation and lifestyle factors, physical health status, mental health status, well-being, chronic disease, and self-perceived health [1,5,6,7,8,9,10,11,12]. Several potential mechanisms, such as a lack of recreational facilities [13], increased exposure to environmental pollution, violence, stressful life events [14], and decreased access to healthcare [15], may explain the influence of neighbourhood deprivation on health.

Cross-sectional studies have established a positive association between point-in-time indicators of neighbourhood deprivation and poor self-perceived health (SPH) [8,9,10,11,12,16,17]. However, a limitation of these studies is that neighbourhood deprivation exposures were mostly measured cross-sectionally. It is therefore unclear whether changes in the dynamics and sequencing of exposure to neighbourhood deprivation affect the health outcomes of the population over time [2,18,19].

Many people are exposed to multiple neighbourhood environments throughout their lives because residential movement or neighbourhood characteristics may change over time [2,18,20]. It is plausible for people to not develop health problems immediately, but for the effect to appear later in life [2,17,18,21]. Previous studies have been suggested that underestimation of neighbourhood effects is likely to occur if exposures over the residential history are ignored [2,22,23]. Three mechanisms have been proposed in the life course theory to conceptualise the relationship between exposure to neighbourhood deprivation and individual outcomes: duration, timing, and order [21]. Duration refers to the accumulation of exposure over time; timing refers to exposure during a particularly sensitive period; and order refers to the general trend of exposure, whether increasing or decreasing, over time [21]. These mechanisms can affect health outcomes differently, and they should be assessed using different indicators. Nuanced indicators, allowing the modelling of different mechanisms of exposure to neighbourhood deprivation, could reveal important information that was previously neglected for the prevention of health issues [2,24].

To assess the effect of the neighbourhood environment on different dimensions of health, SPH has often been used in Canada [3,8,10,12,16,25]. SPH is a reliable and valid measure of key health outcomes [25,26,27]. In addition to the physiological state, SPH can predict other aspects of health that are difficult to capture clinically, such as the psychological state, lifestyle, and functional limitations [28]. This measure of health is particularly relevant when examining the association between neighbourhood deprivation and health, given the multiple biological and psychosocial pathways potentially linking poor neighbourhoods to disease [28].

This study is part of larger research interested in the creation of indicators of neighbourhood deprivation trajectories (NDTs) for the population of Quebec, Canada. NDTs constitute a measure of long-term exposure to deprivation; for example, it examines the sequence of exposure to greater neighbourhood-level deprivation [24,29]. Sequence analysis based on optimal matching and clustering around theoretical types has been used to construct an indicator of NDTs [29]. This indicator should be useful in epidemiological surveillance in Quebec and can be replicated in different provinces of Canada [29]. This study thus aimed to estimate the association between long-term NDTs and poor SPH status. We hypothesised that when controlling for sociodemographic and health characteristics, long-term NDTs with lower deprivation are associated with lower health ratings.

## 2. Materials and Methods

### 2.1. Study Type and Data Source

This retrospective cohort study used Phase-1 data from the Care Trajectories-Enriched Data (TorSaDE) cohort [30]. In this cohort, Quebec residents participated in at least one of three cycles (2007–2008, 2009–2010 and 2011–2012) of the Canadian Community Health Survey (CCHS). Their responses were linked to 17 years of health administrative data from the Régie d’assurance maladie du Québec (RAMQ) [30]. The RAMQ database provided information on residential locations of the TorSaDE cohort participants. Further details of the TorSaDE cohort can be found elsewhere [30].

The three CCHS cycles among the TorSaDE cohort used an identical sampling design and consistent population representation. The obtained estimates are time estimates and should be interpreted as attributes of the average population residing in Quebec province during 2007–2012 [31].

### 2.2. Sample Selection

This study was conducted with 68,268 participants from Phase 1 of the TorSaDE cohort. Of these, 60,792 individuals agreed to have their data linked. For individuals with more than one CCHS participation, only the most recent participation was retained. Our sample was composed of a subpopulation of adults (aged 18 years and older) with complete residential location information. Participants with missing data on SPH reporting (*n* = 74) and the analytic covariable (*n* = 446) were excluded (Figure 1).

### 2.3. Dependent Variable

SPH is the dependent variable. Self-reported information on how individuals perceive their own general health was obtained by CCHS using the question, “In general, would you say your health is: excellent, very good, good, fair, or poor?” SPH was defined as poor (coded 1 to indicate the category at risk) or good (coded 0 to indicate the reference) if participants responded fair/poor or excellent/very good/good to this question, respectively.

### 2.4. Exposure Variables

NDT was the exposure variable, which was categorised into nine indicators based on existing studies: Privileged Stable, Medium Upward, Privileged Downward, Deprived Upward Extreme, Privileged Downward Extreme, Medium Stable, Deprived Upward, Medium Downward, and Deprived Stable [29]. These trajectories provide information about the order and accumulation of longitudinal exposure to neighbourhoods. Moreover, they are useful to better understand the neighbourhood effects on health outcomes [29].

NDT was measured using the neighbourhood material deprivation index proposed by Pampalon and Raymond [32]. The deprivation index of the respondents’ neighbourhoods of residence was available for the census years between 1999 and 2012. This information was extracted from the provincial health insurance register and was used to create a residential history. For each participant, the residential information was collected 17 years before the date of the administration of the CCHS. A tercile of annual deprivation was assigned to each neighbourhood of residence, and an individual neighbourhood deprivation sequence was created for each participant [29].

Sequence analysis was applied to create an NDT indicator [29]. Individual neighbourhood deprivation sequences over a 17-year period were grouped into nine theoretical classes based on their similarity [29]. To cluster into nine theoretical sequences, a simple typology of tercile of annual deprivation was considered: three stable trajectories and six trajectories with transitions. A complete description of is available elsewhere [29]. Figure 2 illustrate the theoretical sequences of neighborhood deprivation trajectories.

### 2.5. Covariates

Different relevant covariates (age, sex, education, household income, marital status, survey years, life stress, comorbidity, and having a regular doctor) previously identified in Canada [12,27,33] were investigated as potential confounders and included in this study.

Age was operationalised as an ordinal categorical variable (18–39, 40–64, and 65 and over) based on recent work focusing on SPH in Canada [33]. Education was assessed as an ordinal categorical variable: lower than a high school diploma, high secondary school completion, post-secondary completion, and university education. Household income was categorised into the lowest, low middle, middle, high-middle, and highest quintiles. Marital status was assessed using a nominal categorical variable: married/couple, widow/separated/divorced, and single. Immigration status was operationalised as a nominal categorical variable born in Canada, established immigrant (five or more years in Canada), and recent immigrant (less than five years in Canada).

Both life stress measures were self-reported exposures, and participants’ responses were categorised into not at all stressful, not very stressful, slightly stressful, and quite a bit stressful/extremely stressful. The potential for access to primary healthcare was captured through the CCHS question, “Do you have a regular medical doctor?” This question was asked of all respondents who answered as yes or no. Comorbidity was defined as av ≥ 2 conditions measured using the Combined Charlson and Elixhauser Comorbidity index [34]. This index was calculated for the year prior to the CCHS interview. This index represents the sum of weights derived from 30-day mortality predictions for each identified condition, with a higher index indicating higher morbidity [34].

Individuals with missing data for the outcome, exposure, and confounders were excluded from the study sample (Figure 1).

### 2.6. Statistical Analysis

Descriptive statistics were used to calculate frequencies of the analytic variables. The proportion of participants who reported poor SPH (and their confidence interval [CI] at 95%) was estimated for each analytic variable.

The models were constructed using targeted model selection, as suggested by Hosmer et al. [35]. Associations between poor reported SPH and potential confounders were assessed using a logistic regression analysis. Only the variables associated with poor SPH (*p* < 0.20) in the univariate analysis were considered in the multivariate regression.

Multivariate regression models were used to assess the association between poor SPH and long-term NDT, considering potential confounders. Subgroup analyses according to age were also conducted. The contribution of each potential confounder was evaluated in a multivariate model using the Wald test. The odds ratio (OR) and adjusted OR were reported with a 95% CI. The lack of fit for the final model was tested using the Hosmer-Lemeshow goodness-of-fit statistic using the Stata code suggested by Heeringa et al. [36]. Adjusted probabilities, by age subgroup, of individuals reporting poor SPH were predicted marginals derived from the fitted logistic models.

All statistical analyses were performed using Stata 17 SE. To consider the CCHS sampling plan and protect the confidentiality of respondents, all results were computed using bootstraps and sampling weights [37,38].

## 3. Results

### 3.1. Descriptive Statistics for the Analytic Sample

Descriptive statistics for the 45,990 adults selected in the analytic sample are summarised in Table 1. Briefly, of the respondents, 51% were women, 63% were married, and approximately half (48%) were aged 40–64 years. More than a quarter of the respondents reported that they experienced life stress, and more than 20% did not have a regular medical doctor.

Overall, 10.3% (95% CI: 9.9–10.8) of the sample reported poor SPH. This prevalence differed across NDTs, sociodemographic characteristics, and comorbidities (Table 2). Participants who reported poor SPH were more frequent among those aged more than 65 years (21.6%; 95% CI: 20.5–22.7), those with lower than high school diploma (22.7%; 95% CI: 21.5–24.0), those in the lowest income quintile households (26.3%; 95% CI: 24.7–28.0), and those with a comorbidity (33.6%; 95% CI: 31.5–35.8).

In the Privileged Stable Group, only 6.4% (95% CI: 5.7–7.2) of the respondents reported poor SPH (Table 2). The prevalence of poor SPH was highest in the Deprived Stable group (16.4; 95% CI: 15.0–17.8), followed by the Medium Downward group (13.3%; 95% CI: 11.9–14.8), Deprived Upward group (12.1%; 95% CI: 10.8–13.6), and Privileged Downward Extreme group (11.6%; 95% CI: 9.0–14.9).

### 3.2. Association between the Long-Term NDTs and Poor SPH

Univariate logistic regression results showed that age, education, household income, marital status, immigration status, having a medical doctor, comorbidity, and stress were significant predictors associated with poor SPH (Table 2). Sex and year of survey were not associated with poor reported SPH (Table 2).

Adjusted regression models showed that higher odds of poor reported SPH (OR 1.80; 95% CI: 1.50–2.16) were observed in the Deprived Stable trajectories compared with the Privileged Stable trajectories (the most advantaged; Table 3). Respondents categorised as declining from the advantaged trajectory—notably, Privileged Downward (OR: 1.33; 95% CI: 1.09–1.63), Privileged Downward Extreme (OR: 1.63; 95% CI: 1.14–2.32), and Medium Downward (OR: 1.64; 95% CI: 1.35–2.0) trajectories—had higher odds of poor reported SPH compared to people in the Privileged Stable trajectory (Table 3). However, there were no significant differences between respondents categorised into the Deprived Upward Extreme (OR: 1.23; 95% CI: 0.92–1.74) and Privileged Stable trajectories (Table 3).

### 3.3. Subgroup Analyses

Figure 3 presents the predicted probabilities of a respondent reporting poor SPH according to the age subgroup. There were no significant differences in poor health status between the nine NDT groups among those aged under 40 years (Figure 3a).

Among adults aged 40–64 years, the adjusted predicted probabilities of poor SPH were higher for the Deprived Stable (0.12; 95% CI: 0.11–0.14), Medium Downward (0.12; 95% CI: 0.10–0.14), Deprived Upward (0.11; 95% CI: 0.09–0.14), and Privileged Downward Extreme (0.11; 95% CI: 0.09–0.13) groups compared with the Privileged Stable group (0.07; 95% CI: 0.06–0.08). Among those aged over 65 years (0.18; 95% CI: 0.16–0.21), there were no significant differences in poor health status between the Privileged Stable group and the other groups, except for Deprived Stable (0.23; 95% CI: 0.24–0.29).

## 4. Discussion

### 4.1. Key Results

Using a large cohort of the adult population, this study examined variations in the reporting of poor SPH across different types of NDTs and determined whether these variations change according to age groups. These findings extend previous research on neighbourhoods’ effects on health by using long-term NDTs instead of only one time point. The study showed a detrimental association between long-term exposure to disadvantaged neighbourhoods and poor SPH. Participants who were classified as having disadvantaged trajectories (Deprived Stable, Medium Stable, Deprived Upward, Medium Downward, and Deprived Stable) and those who declined to disadvantaged trajectories (Privileged Downward and Privileged Downward Extreme) had higher odds of reporting poor SPH, even after controlling for confounders.

Participants with sustained exposure to neighbourhood deprivation over their life course had higher odds of developing poor SPH. Similar to previous studies, cumulative exposure to greater neighbourhood deprivation was associated with poor SPH [17,21,39]. This finding is consistent with the life course theory, which suggests that accumulation of exposure to neighbourhood deprivation at different life stages may be harmful for SPH [4]. The results of this study support the existing evidence suggesting an association between long-term NDTs and poor health outcomes [2,19,24,40,41,42]. Using NDTs, Letarte et al. found that women exposed to disadvantaged trajectories (Deprived Upward, Average Downward, or Deprived Trajectory) had higher odds of living with obesity than those exposed to the Privileged trajectory [24]. Additionally, studies using similar NDTs found that neighbourhood deprivation was associated with diabetes [7], psychological distress [19], weight gain [40,43] and intimate partner violence [41]. Using trajectories of exposure to high-poverty neighbourhoods, Yang and South found that neighbourhood poverty trajectories were a stronger predictor of self-rated health around age 40 [23]. According to a study by Johnson et al., living in poor neighbourhoods during young adulthood is strongly associated with negative health outcomes later in life [44].

Many plausible pathways explain which long-term NDTs might impact general health status. Previous studies suggested that inhabitants living for long periods in deprived neighbourhoods might be more exposed to environmental injustice, such as high noise, air pollution, poor built environments, and poor housing conditions, than their counterparts in privileged neighbourhoods [45,46]. Compared to their counterparts in more advantaged areas, living a long time in deprived areas might expose individuals to multiple chronic stressors as well as inadequate access to healthy foods, recreational opportunities, and healthcare [19,47,48]. Finally, residents with long-term NDTs are more likely to exhibit psychological stress via perceived neighbourhood-related factors, constant exposure to poverty-related issues, and high levels of crime [33,49].

The age-stratified analysis allowed further examination of the life stages in which the association between NDTs and SPH was the most evident. Significant differences in age groups were also observed. No difference in the predicted probabilities of poor SPH was observed across NDTs among the younger age group (18–39 years). However, among participants aged >40 years, the results suggest that neighbourhood deprivation is an important predictor of SPH. The magnitude of this association was higher in adults aged ≥ 65 years.

Analysis stratified by age group showed that life stage was a critical factor in the association between neighbourhood deprivation and SPH. There are several possible explanations for this observation. Living in deprived neighbourhoods during adulthood is strongly associated with poor health outcomes in later life, independent of individual demographic status [44]. Compared to older people who live in better neighbourhood environments, older adults exposed to long-term neighbourhood deprivation report more physical health problems [50]. Perhaps the cumulative effect of neighbourhood deprivation is only visible after a long period. People who have a stable trajectory (advantaged or disadvantaged) for over 17 years probably have a stable trajectory for even longer.

### 4.2. Implications

Studying the relationship between long-term exposure to neighbourhood deprivation and health status may provide valuable evidence to guide interventions at the neighbourhood level. In agreement with previous research, this study suggested that deterioration in the trajectory of neighbourhood deprivation, caused by movement, change in residential tenure, shifts in economics, politics, and social environments, is associated with adverse health outcomes such as poor SPH, psychological distress, poor life satisfaction, and cardiovascular disease [19,51]. Municipalities in Quebec should acknowledge this and focus on improving conditions that influence health in disadvantaged neighbourhoods.

To appropriately inform decision-makers when developing neighbourhood-based interventions, further research should examine the underlying mechanisms (e.g., healthcare uptake and access, environmental stressors, education, and employment) driving the observed association between NDTs and health status.

### 4.3. Strengths and Limitations

To the best of our knowledge, this is the first study to investigate the association between long-term NDTs and poor SPH. One of the strengths of this study is the use of a novel cohort in Quebec, which linked the CCHS participants to administrative data. This cohort provided a large dataset representative of the Quebec population, which enhanced its statistical power. In addition, information about 17 years of residential history was available, which allowed for the classification of Quebec’s population according to its NDTs. By creating several types of stable, ascending, and declining NDTs, this study provided precise information about the type of trajectory that might be the most detrimental to health status.

Some limitations should be considered for better interpretation of the results. First, this study was restricted to individuals with complete 17-year residential address history. This restriction may have affected the representativeness of the study sample. It is therefore possible that the results underestimate the risk of neighbourhood changes to vulnerable populations, such as migrants, homelessness, and people without stable housing. Second, the NDTs of study participants can begin at different age periods, from childhood to late adulthood. It is possible that adjustment of the study analyses for age is not sufficient to minimise the differential life-stage effect of exposure to neighbourhood deprivation on health status [24]. Third, because SPH was evaluated at a single time point, the observed associations may be biased by reverse causality and neighbourhood self-selection; the latter was found to be a minor source of bias in health studies [52]. Lastly, even if SPH has been used extensively in the health inequality literature, it is considered a limited indicator of population health compared with objective health measures [53].

## 5. Conclusions

This study revealed that NDTs are associated with poor health status, especially among adults aged over 40 years. This finding contributes to the increasing evidence of an association between neighbourhood disadvantage and unhealthy health. Our results suggest the importance of developing sustainable neighbourhood interventions to address health disparities.

## Figures and Tables

**Figure 1 ijerph-20-00486-f001:**
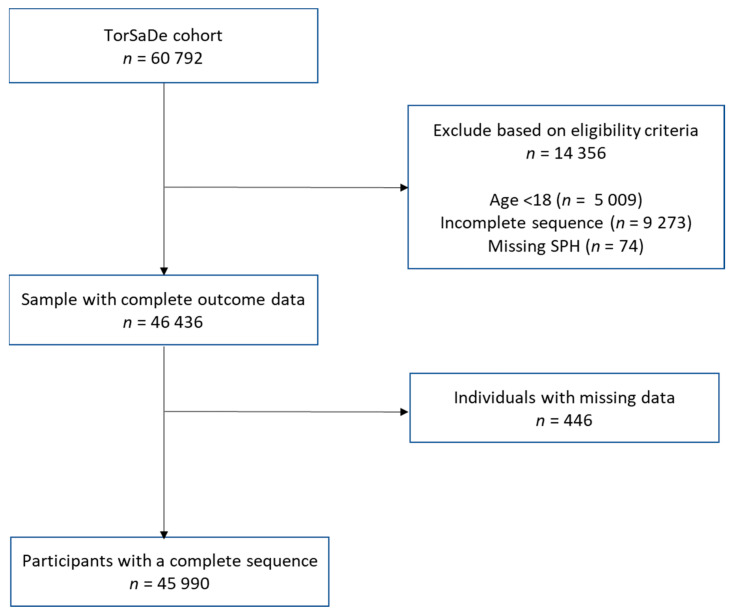
Exclusion flow chart of analytic sample.

**Figure 2 ijerph-20-00486-f002:**
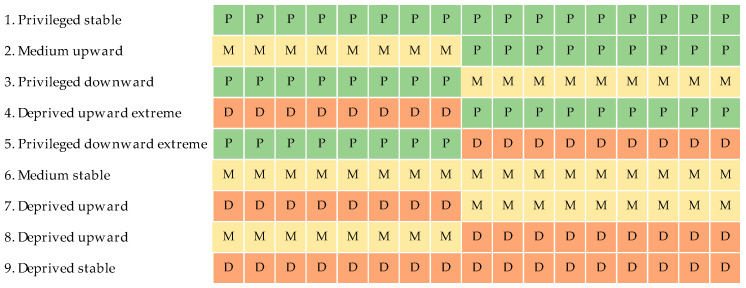
Theoretical sequences of neighborhood deprivation trajectories based on the tercile of annual deprivation. Notes: Privileged (P), Medium (M), Deprived (D).

**Figure 3 ijerph-20-00486-f003:**
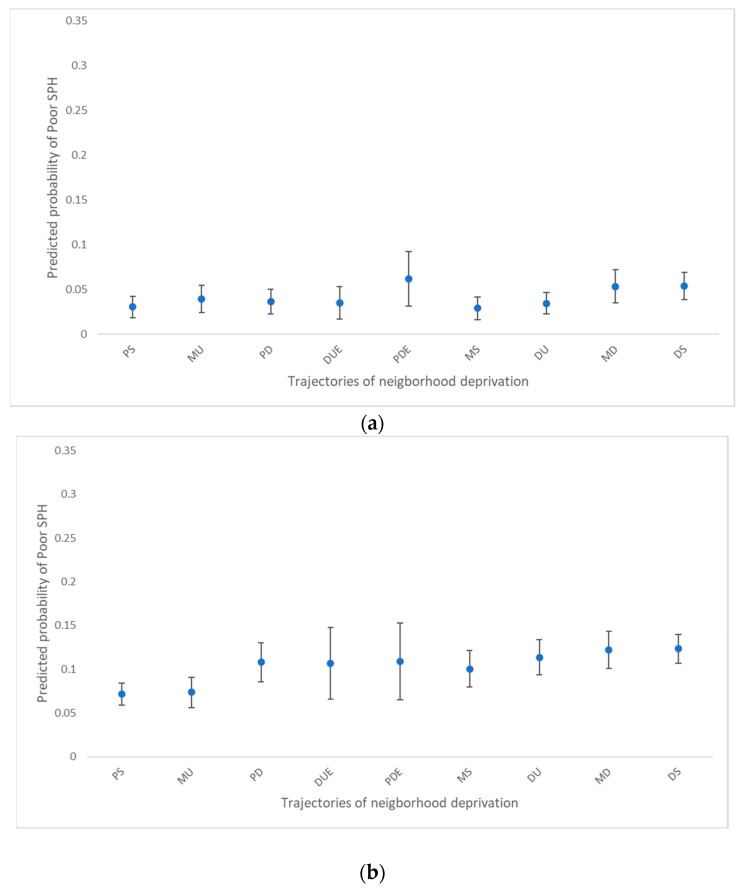
Predicted probability of poor SPH over neighbourhood deprivation trajectories (NDT) by age group (**a**) under 40 years, (**b**) 40–64 years, (**c**) over 65 years. Notes: predicted probability of SPH is shown on the *y*-axis and deprivation trajectories group is shown on the *x*-axis. Error bars on graphs indicate 95% confidence intervals. Privileged Stable (PS), Medium Upward (MU), Privileged Downward (PD), Deprived Upward Extreme (DUE), Privileged Downward Extreme (PDE), Medium Stable (MS), Deprived Upward (DU), Medium Downward (MD), Deprived Stable (DS).

**Table 1 ijerph-20-00486-t001:** Demographic characteristics of study sample (*n* = 45,990).

Characteristics	Frequency (%) *
Sex	
Women	51.1
Men	48.9
Age group, years	
18–39	33.1
40–64	48,1
65+	18.8
Education level	
No high school diploma	18.9
High school diploma	13.5
Post-secondary diploma	48.6
University diploma	18.9
Household income quintile	
Q1 (Low SES)	9.3
Q2	20.9
Q3	20.0
Q4	16.9
Q5 (High SES)	33.0
Marital status	
Married–Couple	62.6
Widow–Separated–Divorced	14.3
Single	23.0
Immigration status	
Immigrant	8.0
Non-immigrant	92.0
Survey cycle	
2007–2008	33.3
2009–2010	33.7
2011–2012	33.0
Life stress	
None/low stress	73.0
High stress	27.0
Comorbidity	
0	90.5
≥1	9.5
Have a regular doctor	
Yes	77.8
No	22.2
Trajectory of neighbourhood deprivation	
Privileged stable	21.7
Medium upward	10.4
Privileged downward	12.1
Deprived upward extreme	2.8
Privileged downward extreme	2.6
Medium stable	14.8
Deprived upward	8.9
Medium downward	11.2
Deprived stable	15.5

* Percentages weighted to the Canadian population.

**Table 2 ijerph-20-00486-t002:** Bivariate associations between poor self-perceived health and independent variables.

Variables	Poor SPH * (%)	Unadjusted OR (95% CI)	*p*-Value
**Exposure**			
Trajectory of neighbourhood deprivation			
Privileged stable	6.4 (5.7–7.2)	1	<0.001
Medium upward	7.2 (6.2–8.3)	1.13 (0.92–1.37)	
Privileged downward	9.1 (8.0–10.5)	**1.47** (1.22–1.76)	
Deprived upward extreme	7.8 (6.1–9.9)	1.23 (0.92–1.64)	
Privileged downward extreme	11.6 (9.0–14.9)	**1.91** (1.37–2.67)	
Medium stable	9.9 (8.7–11.2)	**1.60** (1.34–1.91)	
Deprived upward	12.1 (10.8–13.6)	**2.01** (1.68–2.41)	
Medium downward	13.3 (11.9–14.8)	**2.24** (1.88–2.66)	
Deprived stable	16.4 (15.0–17.8)	**2.85** (2.44–3.34)	
**Potential confounding**			
Sex			0.58
Women	10.5 (9.9–11.1)	1	
Men	10.2 (9.6–10.9)	0.97 (0.88–1.07)	
Age group, years			<0.001
18–39	4.0 (3.6–4.6)	1	
40–64	10.3 (9.6–11.0)	2.71 (2.31–3.22)	
65+	21.6 (20.5–22.7)	6.53 (5.67–7.50)	
Education level			<0.001
No high school diploma	22.7 (21.5–24.0)	6.04 (4.95–7.38)	
High school diploma	10.4 (9.1–11.8)	2.38 (1.90–2.99)	
Post-secondary diploma	7.7 (7.2–8.3)	1.72 (1.41–2.10)	
University diploma	4.6 (3.9–5.5)	1	
Household income distribution quintile			<0.001
Q1 (Low SES)	26.3 (24.7–28.0)	7.50 (6.24–9.00)	
Q2	16.8 (15.7–18.1)	4.25 (3.56–5.08)	
Q3	9.2 (8.3–10.2)	2.14 (1.78–2.58)	
Q4	6.0 (5.3–6.9)	1.35 (1.08–1.67)	
Q5 (High SES)	4.5 (3.9–5.3)	1	
Marital status			<0.001
Married–Couple	9.2 (8.7–9.8)	1	
Widow–Separated–Divorced	18.8 (17.4–20.4)	2.29 (2.03–2.57)	
Single	8.1 (7.4–8.9)	0.87 (0.77–0.98)	
Immigration status			<0.001
Immigrant	9.9 (9.5–10.3)	1.67 (1.37–2.02)	
Non-immigrant	15.5 (13.2–18.1)	1	
Survey cycle			0.58
2007–2008	10.4 (9.7–11.1)	1	
2009–2010	10.1 (9.4–10.7)	0.97 (087–1.07	
2011–2012	10.6 (9.8–11.4)	1.02 (0.92–1.14)	
Life stress			<0.001
None/low stress	9.8 (9.3–10.4)	1	
High stress	11.7 (10.8–12.6)	1.21 (1.10–1.34)	
Comorbidity			<0.001
0	7.9 (7.5–8.3)	1	
≥1	33.6 (31.5–35.8)	5.93 (5.31–6.61)	
Have a regular doctor			<0.001
Yes	11.8 (11.2–12.3)	2.37 (2.04–2.76)	
No	5.3 (4.7–6.1)	1	

* Percentages weighted to the Canadian population. Bold values indicate statistical significance.

**Table 3 ijerph-20-00486-t003:** Associations between long-term NDT and poor self-perceived health.

Variables	Adjusted OR (95% CI)	*p*-Value
**Exposure**		
Trajectory of neighbourhood deprivation		<0.001
Privileged stable	1	
Medium upward	1.11 (0.89–1.37)	
Privileged downward	**1.33** (1.09–1.63)	
Deprived upward extreme	1.27 (0.92–1.74)	
Privileged downward extreme	**1.63** (1.14–2.32)	
Medium stable	**1.24** (1.00–1.53)	
Deprived upward	**1.50** (1.22–1.83)	
Medium downward	**1.64** (1.35–2.00)	
Deprived stable	**1.80** (1.50–2.16)	
**Potential confounding**		
Age group, years		<0.001
18–39	1	
40–64	2.25 (1.90–2.65)	
65+	2.93 (2.46–3.50)	
Education level		<0.001
No high school diploma	2.68 (2.14–3.36)	
High school diploma	1.71(1.33–2.20)	
Post-secondary diploma	1.39 (1.13–1.71)	
University diploma	1	
Household income distribution quintile		<0.001
Q1 (Low SES)	3.61 (2.85–4.58)	
Q2	2.27 (1.83–2.81)	
Q3	1.56 (1.26–1.92)	
Q4	1.12 (0.89–1.41)	
Q5 (High SES)	1	
Marital status		0.15
Married–Couple	1	
Widow–Separated–Divorced	1.08 (0.93–1.25)	
Single	1.15 (1.00–1.34)	
Immigration status		<0.001
Immigrant	1.52 (1.23–1.87)	
Non-immigrant	1	
Life stress		<0.001
None/low stress	1	
High stress	2.07 (1.82–2.34)	
Comorbidity		<0.001
0	1	
≥1	3.92 (3.45–4.46)	
Have a regular doctor		<0.001
Yes	1.54 (1.30–1.81)	
No	1	

Bold values indicate statistical significance.

## Data Availability

The data used in this study are publicly available from the Institut de la statistique du Québec. Some restrictions apply to the availability of these data, which were used under license for the current study, and so are not publicly available. Data are however available from the authors upon reasonable request and with permission of ISQ, authorization from the CAI, and other appropriate approvals from relevant data stewards.

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
