# Peer review of "Association between Neighbourhood Deprivation Trajectories and Self-Perceived Health: Analysis of a Linked Survey and Health Administrative Data"

_ijerph, 2022, doi:10.3390/ijerph20010486_

Round 1

Reviewer 1 Report

Overall, this is a well-written manuscript understanding how neighbourhood deprivation trajectories are associated with self-perceived health. I only have a few comments as follows.

- In this study, "Quebec residents participated in at least 79 one of four cycles (2007–08, 2009–10, 2011–12, and 2013–14) of the Canadian Community 80 Health Survey (CCHS)." For participants with more than one cycle participation, which cycle did the authors select in their study?

- From the manuscript, 17-year residential information represents the period from 1999 to 2016 and it overlapped with the period of residents' participation in CCHS (2007-2014). I was wondering why the authors did not use a shorter period of residential information, particularly up to or before the participation in CCHS. 

- It would be better if the authors could add a figure that illustrates nine different deprivation trajectories, and explain in a few sentences on sequence analysis used to develop the trajectories.

- The analyses should have used "multilevel" logistic regression models since the exposure was assessed at the neighbourhood level. It is important to correctly estimate the standard error. 

Author Response

Reviewer 1

Overall, this is a well-written manuscript understanding how neighbourhood deprivation trajectories are associated with self-perceived health. I only have a few comments as follows.

Comment 1: In this study, "Quebec residents participated in at least 79 one of four cycles (2007–08, 2009–10, 2011–12, and 2013–14) of the Canadian Community 80 Health Survey (CCHS)." For participants with more than one cycle participation, which cycle did the authors select in their study?

Response: Thank for your comment. For individuals with more than one CCHS entry, only the most recent entry was retained. Following your comment, we have added this information in the manuscript. Please see Lines 91-92:  

“For individuals with more than one CCHS participation, only the most recent participation was retained”

Comment 2: From the manuscript, 17-year residential information represents the period from 1999 to 2016 and it overlapped with the period of residents' participation in CCHS (2007-2014). I was wondering why the authors did not use a shorter period of residential information, particularly up to or before the participation in CCHS. 

Response: Thank you for this comment. We have used long-term exposure to neighbourhood deprivation because previous studies suggest that underestimation of neighborhood effects is likely to occur if exposures over the residential history are ignored (Clarke et al., 2014; Do, 2009; Yang and South, 2018). According to life course theory, accumulation of exposure to deprivation over time or timing of exposure to deprivation is associated with adverse health outcome. For each participant, the residential information was collected for up to 17-years before the index date in the CCHS. The index date in this study is the date of the administration of the CCHS to each participant. This date is between 2007 and 2012 depending on the cycle in which the participants were enrolled. We have made the necessary change in the manuscript and included the study period. Please, see:

lines 47-49 - “Previous studies suggest that underestimation of neighborhood effects is likely to occur if exposures over the residential history are ignored [].

lines 118-119  “For each participant, the residential information was collected 17 years before the date of the administration of the CCHS”.

Comment 3: It would be better if the authors could add a figure that illustrates nine different deprivation trajectories, and explain in a few sentences on sequence analysis used to develop the trajectories.

Response: Thank you for your comment. A figure 2 illustrate the neighborhood trajectories has been added (Please see Annex 1). This process of sequences analysis was clarified in the manuscript. Please see line 124 - 127.

“To cluster into nine theoretical sequences, a simple typology of tercile of annual deprivation were considered: three stable trajectories and six trajectories with transitions. A complete description of is available elsewhere [27]. Figure 2 illustrate the theoretical sequences of neighborhood deprivation trajectories”

     Figure 2. Theoretical sequences of neighborhood deprivation trajectories based on the tercile of annual deprivation.

Notes: Privileged (P), Medium (M), Deprived (D)

Comment 4: The analyses should have used "multilevel" logistic regression models since the exposure was assessed at the neighbourhood level. It is important to correctly estimate the standard error. 

Response: Thank you for this comment. In our study, the neighborhood deprivation trajectories as exposure variable were estimated at individual level. We have transformed individual residential histories into neighborhood deprivation trajectories. For this reason, we have not a multilevel data structure necessary to realize multilevel analysis. We used svyset procedures, recommended by Statistics Canada’s guidelines, to apply design and bootstrap weights to produce unbiased estimates with variances adjusted for the sampling method.

Reviewer 2 Report

General Comments The article addresses an important issue in understanding to estimate the association between long-term NDTs and poor SPH status. Relevant aspect for health policy.

Introduction:The theoretical framework seems adequate and sufficient.

Materials and Methods: The methods used are explained: choice and selection of the sample, time horizon, measurement instruments, data sources, data analysis methods However, the categorization of the dependent variable should be better clarified: measurement scale, reliability, validity. As well as clarifying how your categorization was done to enter logistic regression models.

Results: Appear clear and adequate, with stratified analysis.

Discussion: The authors acknowledge the limitations of the study and thus the implications for health policy. The authors compare their results with those of other studies in the area, highlighting its distinctive aspect.

Author Response

Reviewer 2

General Comments The article addresses an important issue in understanding to estimate the association between long-term NDTs and poor SPH status. Relevant aspect for health policy.

Introduction: The theoretical framework seems adequate and sufficient.

Materials and Methods: The methods used are explained: choice and selection of the sample, time horizon, measurement instruments, data sources, data analysis methods However, the categorization of the dependent variable should be better clarified: measurement scale, reliability, validity. As well as clarifying how your categorization was done to enter logistic regression models.

Results: Appear clear and adequate, with stratified analysis.

Discussion: The authors acknowledge the limitations of the study and thus the implications for health policy. The authors compare their results with those of other studies in the area, highlighting its distinctive aspect.

Comment 4: "However, the categorization of the dependent variable should be better clarified…". 

Response: We would like to thank Reviewer 2 for the valuable time taken to read our manuscript. Following your comment, we have clarified the categorization of the dependent variable. Please see Lines 103-105.

“SPH was defined as poor (coded 1 to indicate the category at risk) or good (coded 0 to indicate the reference) if participants responded fair/poor or excellent/very good/good to this question, respectively”.